# *ALDH2* p.E504K Variation and Sex Are Major Factors Associated with Current and Quitting Alcohol Drinking in Japanese Oldest Old

**DOI:** 10.3390/genes12060799

**Published:** 2021-05-24

**Authors:** Takashi Sasaki, Yoshinori Nishimoto, Takumi Hirata, Yukiko Abe, Toru Takebayashi, Yasumichi Arai

**Affiliations:** 1Center for Supercentenarian Medical Research, Keio University School of Medicine, 35 Shinanomachi, Shinjuku-ku, Tokyo 160-8582, Japan; ynishimo@keio.jp (Y.N.); tkm2006diabetes@yahoo.co.jp (T.H.); yukiko_abe@keio.jp (Y.A.); yasumich@keio.jp (Y.A.); 2Department of Neurology, Keio University School of Medicine, Tokyo 160-8582, Japan; 3Department of Public Health, Faculty of Medicine, Hokkaido University, Sapporo 060-8638, Japan; 4Department of Preventive Medicine and Public Health, Keio University School of Medicine, Tokyo 160-8582, Japan; ttakebayashi@a3.keio.jp

**Keywords:** alcohol, current drinking, Oldest Old, ALDH2, rs671

## Abstract

This study identified the factors associated with current and quitting alcohol drinking in the Oldest Old to better understand the associated factors and mechanisms underlying drinking behaviors in this age group. Results of a questionnaire for drinking behavior in 1015 Japanese Oldest Old citizens aged 85 to 89 years revealed that 56.0% of men and 24.0% of women were current drinkers. A genome-wide association study revealed that the rs671 G > A variation, which corresponds to the aldehyde dehydrogenase 2 (*ALDH2*) p.E504K missense variant, was significantly associated with current drinking (odds ratio: 3.8, *p* = 3.33 × 10^−31^). Variable selection with 41 factors and multivariate regression logistic analysis for current drinking indicated that the rs671 genotype and sex were the most significant factors in the Oldest Old. Further analysis revealed that the rs671 genotype, alcohol-associated biomarkers, a history of heart or kidney disease, and frailty score are factors associated with quitting drinking in the Oldest Old men, whereas smoking history, walking time, and depression score were factors associated with quitting drinking in the Oldest Old women. These results indicate that the *ALDH2* p.E504K variation is a major factor associated with current and quitting drinking in the Japanese Oldest Old.

## 1. Introduction

Alcohol drinking (drinking) and its effects have been present in societies throughout human history. Social drinking with low to moderate alcohol consumption is thought to be a pleasurable behavior; however, habitual drinking with excess alcohol consumption results in several negative outcomes; it is a risk factor for poor health, road incidents, and alcohol dependence. A global status report on alcohol and health by the World Health Organization (WHO) revealed that 57% (3.1 billion people) of the global population aged 15 years and over had abstained from drinking in the previous 12 months, and 2.3 billion people were current drinkers in 2016 [1]. Alcohol is consumed by more than half of the population in only three world regions, the Americas, Europe, and the Western Pacific, indicating that drinking is biased across regions and civilizations worldwide. In Japan, 76.4% of men and 61.9% of women aged 15 years and over consumed alcohol in 2009, ranking 22nd for men and 30th for women among drinking countries and highest in East Asia [2].

The age of drinking is associated with an important health issue. Globally, 26.5% of adolescents aged 15–19 years are current drinkers, accounting for 155 million adolescents [1,3]. Alcohol consumption affects cognitive decline, cardiovascular health, and brain plasticity in aging, indicating that drinking behavior in older people is an important issue for maintaining good health in old age [4]. Aging is also known to affect drinking behavior. A report by the Alcohol and Ageing Working Group published by Health Scotland revealed that older people are more likely to be “never drinking” and “almost daily drinking,” indicating that frequency of drinking is more “polarized” among older people [5]. A similar increase in non-drinkers among older people has been observed in Japan. At 50 years of age, approximately 90% of Japanese men and 70% of Japanese women drink alcohol at least once a year; however, the frequency of current drinkers declined with age. At the age of 80 years, the frequency of current drinkers declined to approximately 60% of men and 10% of women, indicating that aging is one of the factors associated with quitting drinking [6]. To prevent alcohol-related health problems while maintaining the benefits of alcohol consumption, it is vital to identify both the genetic and non-genetic determinants of drinking behavior in older people. Alcohol dehydrogenase (ADH) and aldehyde dehydrogenase (ALDH) are well-known enzymes involved in alcohol metabolism.

ADH catalyzes the conversion of ethanol into acetaldehyde, the first step in the oxidative metabolism of ethanol, and *ADH1B* is the major gene responsible for alcohol metabolism. The ADH1B protein with Arg48 variation oxidizes ethanol approximately 70- to 80-fold slower than the ADH1B protein with His48 variation, resulting in the slow clearance of alcohol from the blood and a tendency to become alcoholic in the *ADH1B* p.H48R variation carriers [7]. ALDH catalyzes the conversion of toxic acetaldehyde into non-toxic acetic acid, an important detoxification step in the oxidative metabolism of ethanol. The minor allele (A) of *ALDH2* p.E504K variation results in the slow clearance of acetaldehyde from the blood, causing alcohol-related flushing, and hangover [8,9]. These phenotypic differences in *ADH1B* and *ALDH2* variations may explain the different associations for alcohol metabolism. Variants of these alcohol metabolism associated genes have been also reported. The association between the *ALDH2* p.E504K variation, drinking behavior, and amount of alcohol consumption has been well replicated in the Japanese population [10,11,12,13]. Genome-wide association study (GWAS) results revealed that both *ADH1B* p.H48R and *ALDH2* p.E504K variants are associated with alcohol dependency in Chinese people; however, only the *ALDH2* p.E504K variant is associated with drinking behavior in Japanese people [12,14]. In male Han Chinese, the *ADH1B* p.H48R variation was positively associated with alcohol dependency; however, the ALDH2 p.E504K variation was negatively associated with alcohol dependency [14]. These results indicate that *ADH1B* p.H48R and *ALDH2* p. E504K variations are expected to be one of the candidate factors associated with drinking behavior in the Oldest Old; thus, this study included these genetic factors for further association analysis of drinking behaviors in the Oldest Old.

We recently established a cohort for the Oldest Old, the Kawasaki Aging and Wellbeing Project (KAWP), which is a longitudinal cohort study of older people aged between 85 and 89 years without physical disability at baseline [15]. In the KAWP, the data were collected via questionnaires of drinking behavior, including current drinking, drinking amount, age at quitting drinking, and other factors expected to be associated with drinking behavior including medical history and biomarkers based on blood tests. Furthermore, we analyzed genotypes of the KAWP participants using the Infinium Asian Screening Array, which can determine the genotype of 0.65 million single nucleotide variants (SNVs). In this study, we first identified SNVs associated with current drinking in the Japanese Oldest Old, and then, 41 factors including life status, cognitive and mental statuses, disease histories, measurement of physical function, plasma biomarkers, and genetic factors were analyzed by the least absolute shrinkage and selection operator (LASSO) to select the appropriate factors and avoid multicollinearity [16]. Finally, we identified the factors associated with drinking behaviors for current drinking and quitting drinking in the Japanese Oldest Old by multivariate regression logistic analysis using the selected factors to understand the detailed mechanisms of quitting drinking in older people.

## 2. Materials and Methods

### 2.1. Study Population

This study used data from cohort studies of the KAWP [15]. The inclusion criteria of the KAWP were as follows: (1) individuals aged between 85–89 and being a resident of Kawasaki, a city with a population of 1.5 million in the Kanagawa prefecture; (2) individuals having no limitations in the basic activities of daily living; and (3) individuals being able to visit the hospital for cohort survey by themselves. A total of 1026 independent older adults were enrolled in the KAWP and a comprehensive baseline assessment was done, including questionnaires for drinking behavior and life status, cognitive and mental status, disease histories, measurement of physical function, and plasma biomarkers (Table 1). Among these participants, 1015 Oldest Old persons (512 men and 503 women) who responded to the questionnaire regarding drinking behavior and accepted genome DNA sequence analysis were included in this study. The KAWP was approved by the ethics committee of the Keio University School of Medicine (ID: 20160297) and was registered in the University Hospital Medical Information Network Clinical Trial Registry as an observational study (ID: UMIN000026053).

### 2.2. Baseline Characteristics

Methods for collecting baseline characteristics of KAWP including the questionnaire for life status and disease histories, measurement of cognitive and mental statuses, measurement of physical functions, and measurement of biomarkers in plasma have been described previously [15]. Briefly, the questionnaire and health data were obtained from the KAWP baseline survey conducted between March 2017 and December 2018. All participants were invited to hospitals, interviewed, and examined using a study protocol that was designed by the Tokyo Oldest Old Survey on Total Health and Japan Semi-supercentenarian Study, which were previously managed by the Center for Supercentenarian Medical Research, Keio University School of Medicine [17,18].

### 2.3. Genotyping

Total genomic DNA was extracted from whole blood using a FlexGene DNA Kit (QIAGEN, Hilden, Germany). The genotypes of 0.65M SNVs were determined using the Infinium Asian Screening Array-24 v1.0 BeadChip kit, according to the manufacturer’s standard protocol. All microarray scan images were analyzed and then the SNV genotypes were determined using GenomeStudio (version 2013, Illumina, San Diego, CA, USA). Quality filters for both samples and SNV genotypes were also applied by GenomeStudio. No sample was filtered using an SNV call rate filter (cutoff < 98.0, lowest call rate: 0.9803). In addition, 15,781 out of 657,060 SNVs (2.4%) were filtered out by SNV genotyping quality control filters (R Mean (AA, AB, BB) < 0.2500 and Cluster Seq < 0.4000). Finally, the genotypes of 641,279 SNVs from 1016 KAWP participants were obtained.

### 2.4. Pre-Phasing and Imputation

Genotype data on the Infinium Asian Screening Array were converted to VCF format by GenomeStudio (version 2013). These genotype data were prephased by shapeit4.0 program according to documentation with default parameters and genetic map sets (genetic_maps.b37) on the shapit4 web site (https://odelaneau.github.io/shapeit4/, accessed on 5 February 2021) [19]. The SNVs were imputed using prephased genotype data using the impute5 program (version1.1.4, https://jmarchini.org/impute5/, accessed on 5 February 2021) with default parameters and the phased VCF data on the 1000Genomes phase 3 data (ftp://ftp.1000genomes.ebi.ac.uk/vol1/ftp/release/20130502/, accessed on 5 February 2021) [20]. To evaluate the probability of accuracy for imputed variants, “info” metric values calculated by the impute5 program were employed against each imputed variant, and 0.5 was used as a threshold for the impute5 info value as recommended on the web site. Among 80.9 million SNVs imputed by imputed5, 6.38 million SNVs were screened using combination filters for the imputation reliability filter (impute5 info ≥ 0.5), minor allele frequency ≥ 0.02, and *p*-value for Hardy–Weinberg equilibrium >10^−6^ as high-quality SNV data.

### 2.5. Genome-Wide Association Study

To identify SNVs associated with drinking behavior in the Oldest Old, this study analyzed the genome-wide association between the answers for the drinking behavior questionnaire. GWAS was performed using linear models with sex information as covariate by the PLINK program (version 1.90) (version 1.90) [21]. The genomic inflation est. lambda (based on median chisq) was 1.014, indicating that no evident inflation occurred.

### 2.6. Statistical Analysis

The difference in baseline data was evaluated using the *t*-test, Wilcoxon rank-sum test, or chi-square test (Table 1). LASSO is a penalized technique for variable selection that is effective when the number of events per variable is low [16]. LASSO shrinks the coefficients for weaker predictors to zero. The degree of shrinkage was determined by the optimal parameter lambda, as identified by five-fold cross-validation. Multivariate logistic regression analyses were performed using the generalized linear model using factors selected by LASSO. All statistical analyses were performed using the R script (version 4.0.3) with effsize (cohen.d function for effect size estimation of *t*-test data (version 0.8.1)), vcd (assocstats function for effect size estimation of Chi-square test data (version 1.4-8), stat (glm function for multivariate regression analysis (version 4.0.3)), glmnet (glmnet function for LASSO (version 4.1)), exactRankTests (wilcox.exact function for Wilcoxon rank-sum test (version 0.8-31)), rstatix (wilcox_effsize function for effect size estimation of wilcoxon rank-test data (version 0.7.0)) and default packages.

## 3. Results

### 3.1. Baseline Characteristics of the Kawasaki Aging and Wellbeing Project Cohort

A baseline survey was conducted with the participants of the KAWP, an ongoing longitudinal cohort study of older adults aged 85 years or older [15]. The participants’ characteristics are presented in Table 1. The total number of participants with genomic data analyzed through DNA microarray was 1015, of which 287 men (56.0%) and 121 women (24.0%) were current drinkers (Figure 1a). 

### 3.2. Imputation and GWAS for Current Drinking in Japanese Oldest Old

To detect genetic variants associated with current drinking in the Oldest Old, this study analyzed SNVs by a GWAS with a two-choice questionnaire (yes/no) for current drinking, using the imputed DNA sequences of Japanese Oldest Old (Figure 1b). The 0.65 million SNVs were genotyped using the Infinium Asian Screening Array against the 1016 Oldest Old. These SNV data were prephased and then imputed using 1000Genomes phase 3 variation data. After filtering using the imputation reliability, allele frequency, and Hardy–Weinberg equilibrium filters, the genotypes of 6.38 million imputed SNVs were obtained. We then analyzed the association between the two-choice questionnaire for current drinking and genotypes in 1015 older adults using the plink 1.90 program. As a result, rs671 G > A SNV, which corresponds to the *ALDH2* p.E504K missense variant, was detected as the most significant variant (odds ratio: 3.8, *p* = 3.33 × 10^−31^, adjusted *P* (Bonferroni) = 2.17 × 10^−24^) associated with current drinking in the Japanese Oldest Old.

### 3.3. ALDH2 p.E504K Variation Is Associated with Drinking Behavior, Frequency, and Amount in the Japanese Oldest Old

To confirm the association between current drinking and ALDH2 p.E504K variation in the Oldest Old, the frequency of current drinkers was compared in each genotype (Figure 2a). The results revealed that 82.8% of 250 Oldest Old men without the *ALDH2* p.E504K variation were current drinkers; however, 36.4% of 220 and 0% of 42 Oldest Old men with the heterozygous and homozygous *ALDH2* p.E504K variation were current drinkers, respectively. In women, 33.7% of 288 Oldest Old women without the *ALDH2* p.E504K variation were current drinkers; however, 12.1% of 190 and 4.0% of 25 Oldest Old women with the heterozygous and homozygous *ALDH2* p.E504K were current drinkers, respectively. These results indicate that the *ALDH2* p.E504K variation is significantly associated with current drinkers among Oldest Old men and women.

Furthermore, this study analyzed the association between drinking frequency and *ALDH2* p.E504K variation in current drinkers (Figure 2b). An individual who drinks three times or more per week was defined as a “frequent drinker.” As a result, 83.1% of 207 current drinkers among the Oldest Old men without the *ALDH2* p.E504K variation is considered as significantly frequent compared to the 63.3% of 79 current drinkers among the Oldest Old men with the *ALDH2* p.E504K variation. This study found a frequency difference between current drinkers among Oldest Old women without *ALDH2* p.E504K variation and those with *ALDH2* p.E504K variation; however, the difference was not significant.

Finally, the association between the drinking amount and *ALDH2* p.E504K variation in frequent and non-frequent drinkers was analyzed in this study (Figure 2c,d). An individual who drinks over 633 mL of beer and less or equal to 633 mL of beer per day was defined as a “heavy drinker” and “light drinker,” respectively. As a result, 33.1% of 172 frequent drinkers among the Oldest Old men without the *ALDH2* p.E504K variation were heavy drinkers and were significantly more frequent than the 12.0% of 50 frequent drinkers among the Oldest Old men with *ALDH2* p.E504K variation. For other combinations, including non-frequent drinkers among the Oldest Old men and both frequent and non-frequent drinkers among the Oldest Old women, no significant difference in alcohol consumption was observed.

### 3.4. Distribution of Age at Quitting Drinking in Oldest Old Men and Women with the ALDH2 p.E504K Variation

To clarify the associations among frequency of quitting drinking, *ALDH2* p.E504K variation, and age at quitting drinking in the Oldest Old, frequency and age distribution of past drinkers were analyzed. In the Oldest Old men, the frequency of past drinkers was significantly different depending on the *ALDH2* p.E504K variation (Figure 3a). Distribution of age at quitting drinking in the Oldest Old men indicated that the frequency of past drinkers without *ALDH2* p.E504K variation mainly started to increase at the age of 60 and continued to increase until the age of 80; however, the frequency of past drinkers with heterozygous *ALDH2* p.E504K variation started to increase in their 50s and peaked in their 60s, indicating that the *ALDH2* p.E504K variation would also affect the age at quitting drinking in Japanese men (Figure 3b). For women, no significant difference was found between the frequency of past drinkers by *ALDH2* p.E504K variation (Figure 3c). The distribution of age at quitting drinking in the Oldest Old women indicated that past drinkers without the *ALDH2* p.E504K variation in women showed a pattern similar to that in men; however, no evident pattern was observed for past drinkers with heterozygous *ALDH2* p.E504K variation in women because there were only eight past drinkers with heterozygous *ALDH2* p.E504K variation (Figure 3d). These results suggest that *ALDH2* p.E504K variation does not affect the age at quitting drinking in Japanese women.

### 3.5. Variable Selection Associated with Current Drinking by LASSO and Its Multivariate Regression Logistic Analysis

For multivariate regression logistic analysis associated with current drinking, 41 factors were selected, including two known drinking behavior-associated genetic factors (genotypes of rs671 in *ALDH2* and rs1229984 in *ADH1B*) and 39 factors from the baseline survey of the KAWP (sex, questionnaire for drinking behavior and life status, cognitive and mental statuses, disease histories, measurement of physical function, and biomarkers in plasma) of the Oldest Old (Table 1). To select the variables for multivariate regression logistic analysis, the LASSO method was employed with five-fold cross-validation against 412 Oldest Old men and 385 Oldest Old women without missing values. As a result of five-fold cross-validation, 23 factors were selected for further multivariate logistic analysis for the Oldest Old (Figure 4a). As a result of multivariate regression logistic analysis with these 23 factors, 8 significant factors, including 4 positively associated factors (Body Mass Index (BMI), high-density lipoprotein cholesterol (HDLC), years of education, and γ glutamyl transpeptidase (γGTP) and 4 negatively associated factors (genotype of rs671 (*ALDH2*), sex (women), alanine transaminase (ALT), and history of renal disease) were found (Figure 4b). We also analyzed men and women separately, and 13 and 5 factors were selected for further multivariate logistic analysis for Oldest Old men and women, respectively (Appendix A). As a result of multivariate regression logistic analysis with these 13 factors for Oldest Old men, nine significant factors, including six positively associated factors (HDLC, smoking history, cholinesterase (CHE), years of education, γ GTP, and BMI) and three negatively associated factors (genotype of rs671 (*ALDH2*), alkaline phosphatase (ALP), and history of heart disease), were found (Appendix A). Furthermore, only one of the five factors examined, rs671 (ALDH2), was significant and negatively associated with current drinking in Oldest Old women (Appendix A).

### 3.6. Variable Selection Associated with Quitting Drinking by LASSO and Its Multivariate Regression Logistic Analysis

For multivariate regression logistic analysis associated with quitting drinking, this study selected 230 current and 81 past drinkers among Oldest Old men and 91 current and 36 past drinkers among Oldest Old women. Moreover, 41 factors were selected, including two known drinking behavior-associated genetic factors (genotypes of rs671 in *ALDH2* and rs1229984 in *ADH1B*) and 39 factors from the baseline survey of the KAWP. As a result of LASSO with five-fold cross-validation, 15 and 4 factors were selected for further multivariate logistic analysis for Oldest Old men and women, respectively (Figure 5a,b). As a result of multivariate regression logistic analysis for Oldest Old men, this study found nine significant factors, including six positively associated factors (genotype of rs671 (*ALDH2*), ALT, ALP, history of heart disease, frailty score, and history of renal disease) and three negatively associated factors (γGTP, CHE, and HDLC; Figure 5a). Additionally, only one factor (past smoking) was found to be a significantly positively associated with these four factors for Oldest Old women, through multivariate regression logistic analysis (Figure 5b).

## 4. Discussion

In this study, we identified the factors associated with both current and quitting drinking against 1015 Japanese aged 85 to 89 years from the KAWP cohort. The questionnaire for drinking behavior against the KAWP cohort revealed that 56.0% of men and 24.0% of women were current drinkers, which is comparable to a previous report on Japanese drinking behavior [6]. A GWAS revealed that the rs671 SNV, which corresponds to the *ALDH2* p.E504K missense variant, is significantly associated with current drinking. Further analysis revealed that the *ALDH2* p.E504K variation is also associated with quitting drinking in Oldest Old men; however, only past smoking was associated with quitting drinking in Oldest Old women. The variable selection by LASSO and further multivariate regression logistic analysis indicated that the genotype of rs671 and sex are significantly associated with current drinking among a comprehensive set of 41 investigated factors in the Oldest Old. These results indicate that the *ALDH2* p.E504K variation is one of the major factors for several drinking behaviors, including current and quitting drinking, even in the Japanese Oldest Old.

Previous GWAS results revealed that both *ADH1B* and *ALDH2* variants are associated with alcohol dependency in Chinese people; however, only the *ALDH2* variant is associated with drinking behavior in Japanese people [12,14]. In this study, GWAS for current drinking revealed that the *ALDH2* p.E504K variation was associated with current drinking even in the Japanese Oldest Old; however, no association was found between current drinking and *ADH1B* variation, which is consistent with previous studies [12]. These results suggest that the *ALDH2* p.E504K variation negatively affects drinking behaviors, including current and quitting drinking, by slowing the clearance of acetaldehyde from the blood.

The results of this study also revealed that the frequency of quitting drinking was significantly associated with *ALDH2* p.E504K variation in the Oldest Old men. Multivariate regression logistic analysis between current and past drinkers among the Oldest Old men revealed that *ALDH2* p.E504K variation, biomarkers associated with drinking, disease histories for heart and kidney, and frailty score were associated with quitting drinking. Among these factors, the frailty score and disease histories for the heart and kidney suggest that Older men with health problems tend to quit drinking. Furthermore, the average eGFR calculated with creatinine in past drinkers without *ALDH2* p.E504K variation (homozygotes of rs761 major allele) in their 70–80s is significantly lower than that in the early quit or current drinkers without *ALDH2* p.E504K variation. These results suggest that health deterioration could be one of the reasons for the increasing number of past drinkers without the *ALDH2* p.E504K variation after their 60s, and appropriate health guidance based on *ALDH2* p.E504K genotype by health care providers could be effective to prevent alcohol related-health problems for the Oldest Old. This study also revealed that the frequency of past drinkers with heterozygous *ALDH2* p.E504K variation started to increase in their 50s and peaked in their 60s. Although there is no confirmatory data to explain the reason for this peak, it was assumed that most Japanese men retired from their jobs in their 60s, and the social behavior changes associated with job retirement could cause a decrease in the chance of drinking and promote quitting drinking in Japan. Another possibility is that a low capability for alcohol/aldehyde degradation by *ALDH2* p.E504K variation caused low alcohol tolerance, resulting in it being easier to quit drinking.

In this study, the cohort data analysis with the DNA microarray data enable us to calculate multivariate regression logistic analysis with both environmental and genetic factors. The combination of cohort and genetic data makes it possible to analyze the gene and environmental interaction. Although only two genetic factors are strongly associated with current drinking, our data revealed that the KAWP dataset is valuable not only as an Oldest Old cohort, but also cohort data in which both environmental and genetic factors are available.

This study has several limitations. First, the size of our cohort was small for GWAS; therefore, our GWAS could only detect strongly-associated genetic factors. Furthermore, it has already been reported that the *ALDH2* p.E504K variation is associated with the risk of esophageal and stomach cancers in East Asia [22]. This study could not find a significant association between *ALDH2* p.E504K variation and esophageal cancer history because only four cases of esophageal cancer were observed in the entire KAWP cohort, indicating that a larger cohort is needed to verify the association between alcohol-related cancers and *ALDH2* p.E504K variation. Second, we could not measure all factors associated with drinking behaviors although we measured a comprehensive set of factors potentially associated with drinking behaviors. Another limitation is that the KAWP cohort participants included relatively healthy older adults; therefore, older adults who have alcohol disorders, dementia, and depression might have been possibly less frequent. In future studies, this cohort will be developed to clarify the association between genotype and these diseases.

## 5. Conclusions

This study analyzed the factors associated with the drinking behavior of the Oldest Old in Japan by GWAS and multivariate regression logistic analysis. It was revealed that *ALDH2* p.E504K variation and sex were the major factors associated with current and quitting drinking in the Japanese Oldest Old. This study also suggested that age at quitting drinking was associated with *ALDH2* p.E504K variation although other environmental factors might also be associated in the Oldest Old.

## Figures and Tables

**Figure 1 genes-12-00799-f001:**
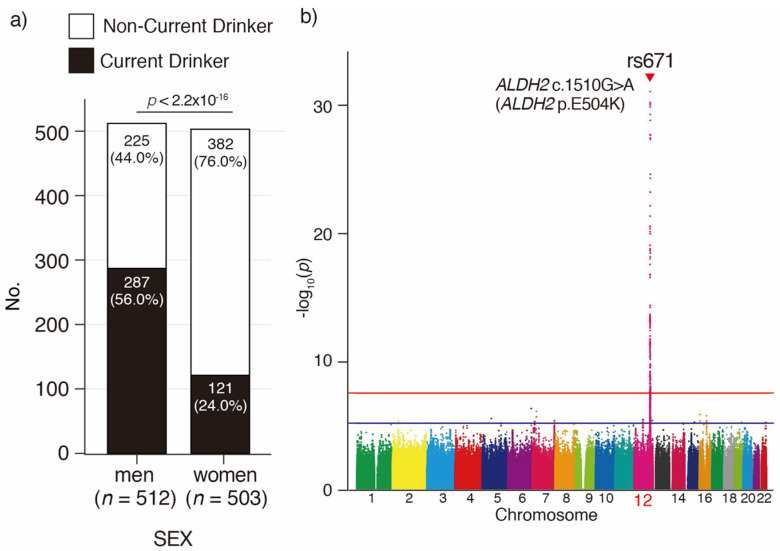
Genome wide association study associated with current drinking in Japanese Oldest Old. (**a**) Number and frequency of current and non-current drinkers in the Japanese Oldest Old. Frequency of current drinkers is significantly different between men and women (chi-square test). (**b**) Genome wide association study associated with current drinking. Red line: 5.0 × 10^−8^, blue line: 1.0 × 10^−5^.

**Figure 2 genes-12-00799-f002:**
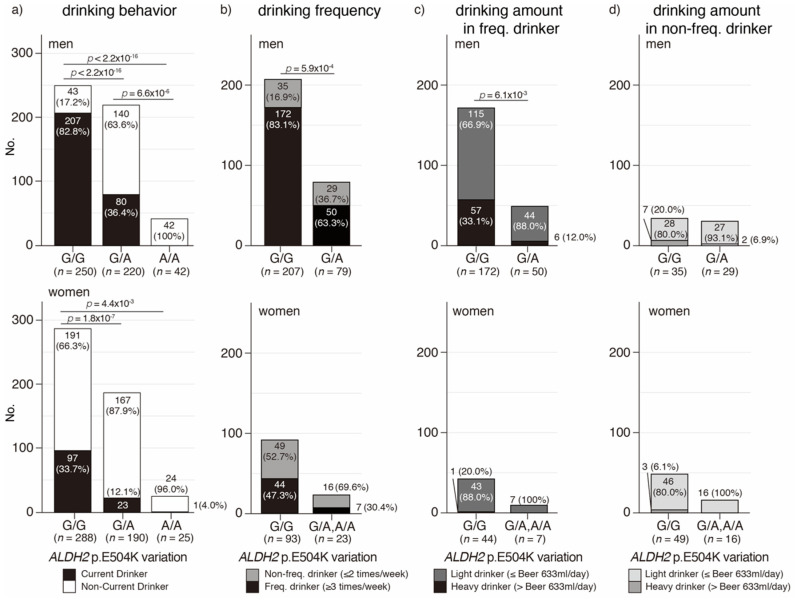
Distribution of Oldest Old for drinking behavior by genotype of ALDH2 p.E504K variation and sex. (**a**) Distribution of current and non-current drinkers by genotype of ALDH2 p.E504K variation. Black: current drinker, white: non-current drinker. (**b**) Distribution of frequent and non-frequent drinkers by genotype of *ALDH2* p.E504K variation. Black: frequent drinker (drinking ≥ 3 times/week); light gray: non-frequent drinker (drinking < 2 times/week). (**c**) Distribution of heavy and light drinkers among frequent drinkers by genotype of *ALDH2* p.E504K variation. Black: heavy drinker (drinking ≥ 633 mL beer/day); dark gray: light drinker (drinking < 633 mL beer/day). (**d**) Distribution of heavy and light drinkers among non-frequent drinkers by genotype of *ALDH2* p.E504K variation. Light gray: heavy drinker (drinking ≥ 633 mL beer/day); dark gray: light drinker (drinking < 633 mL beer/day).

**Figure 3 genes-12-00799-f003:**
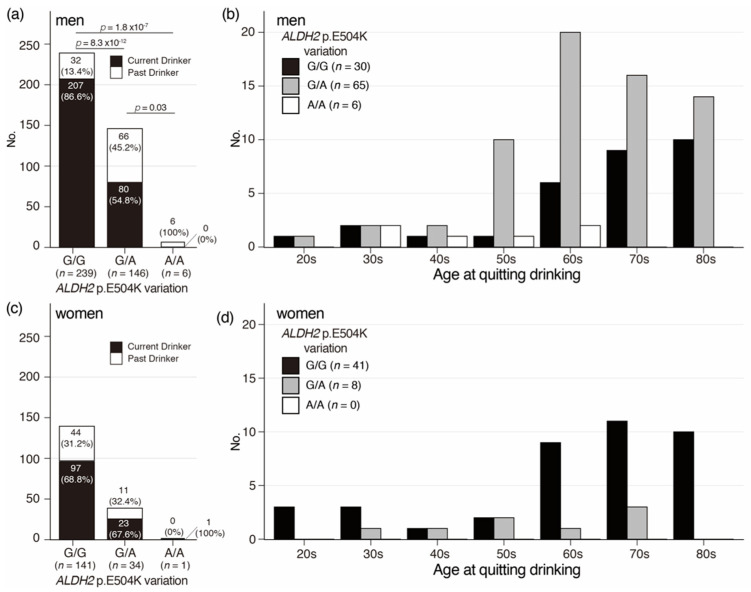
Distribution of Oldest Old for quitting drinking by genotype of *ALDH2* p.E504K variation. (**a**,**b**) Distribution of number of current and former drinkers by genotype of *ALDH2* p.E504K variation in men and women. Black: current drinker, white: former drinker. (**c**,**d**) Distribution of number of Oldest Old for age at quitting drinking in the Oldest Old men and women. Black: *ALDH2* p.E504K non-carrier, gray: *ALDH2* p.E504K heterozygote, white: *ALDH2* p.E504K homozygote.

**Figure 4 genes-12-00799-f004:**
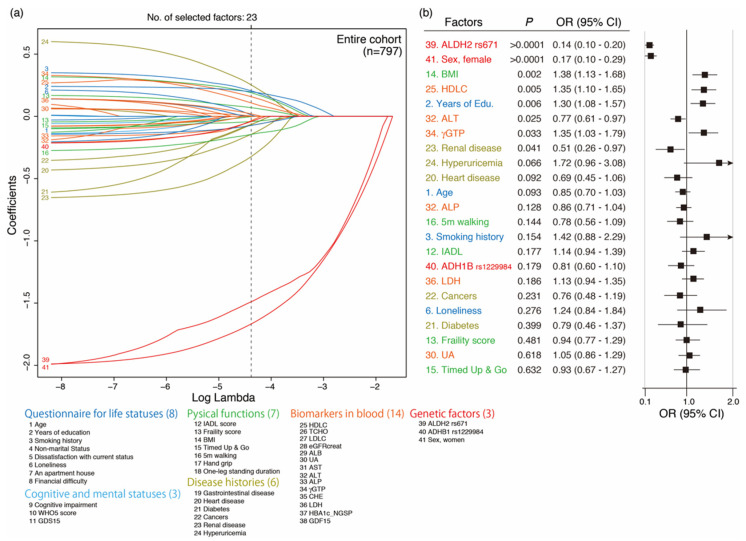
Variable selection associated with current drinking by LASSO and its multivariate regression logistic analysis of Oldest Old. (**a**) Variable selection associated with current drinking by LASSO in Oldest Old men. Five-fold cross-validation to determine an optimal parameter lambda selected 23 factors from 41 factors for further analysis. (**b**) Multivariate regression logistic analysis for current drinking and LASSO selected 8 factors in the Oldest Old.

**Figure 5 genes-12-00799-f005:**
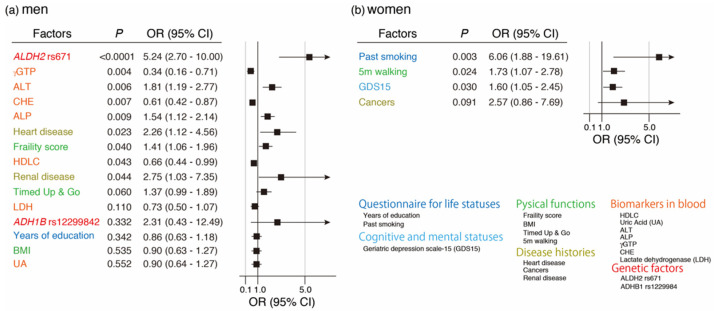
Multivariate regression logistic analysis for quitting drinking against Oldest Old men and women after the variable selection by LASSO (**a**) Multivariate regression logistic analysis for quitting drinking with LASSO selected 15 factors in Oldest Old men (current drinker: *n* = 230, past drinker: *n* = 81). (**b**) Multivariate regression logistic analysis for quitting drinking with LASSO selected 4 factors in Oldest Old women (current drinker: *n* = 91, past drinker: *n* = 36).

**Table 1 genes-12-00799-t001:** Characteristics of KAWP Cohort Research Participants at Enrollment.

			Oldest Old Men					Oldest Old Women				Oldest Old Men vs. Oldest Old Women		
		N	Current Drinker	Non-Current Drinker	*p*	Effect Size ^4^	N	Current Drinker	Non-Current Drinker	*p*	Effect Size ^4^	Men	Women	*p*
No of participants, no		512	287	225	-	-	503	121	382	-	-	512	503	-
No of frequent drinker (≥3 times/week)		286	222	-	-	-	116	51	-	-	-	222	51	<0.001
No of higher amount drinker (>Beer 633ml/time)		286	72	-	-	-	116	4	-	-	-	72	4	<0.001
No of quitted drinker		512	-	96	-	-	503	-	55	-	-	96	55	<0.001
Questionnaire for life statuses (8)														
Age (years), mean (SD) ^1^		512	87.0 (1.3)	87.0 (1.3)	0.645	−0.04	503	86.9 (1.3)	87.1 (1.3)	0.060	−0.20	87.0 (1.4)	87.1 (1.4)	0.330
Educated years (year), mean (SD) ^2^		511	12.3 (3.7)	11.8 (3.5)	0.122	0.07	503	11.0 (2.8)	10.5 (2.4)	0.114	0.07	12.1 (3.7)	10.6 (2.6)	<0.001
Smoking history, no (%) ^3^		512	222 (77.3)	166 (73.7)	0.404	0.04	500	10 (8.3)	29 (7.6)	0.956	0.01	388 (75.8)	39 (7.8)	<0.001
Non-marital Status, no (%) ^3^		509	63 (22.0)	48 (21.6)	0.999	0.01	500	95 (79.8)	295 (77.4)	0.670	0.03	111 (21.9)	390 (78.0)	<0.001
Dissatisfaction with current status, no (%) ^3^		510	45 (15.7)	32 (14.4)	0.787	0.02	501	13 (10.8)	46 (12.1)	0.722	0.02	77 (15.1)	59 (11.8)	0.149
Loneliness, no (%) ^3^		474	91 (33.8)	67 (32.7)	0.870	0.01	470	40 (35.1)	93 (26.1)	0.084	0.09	158 (33.3)	133 (28.3)	0.109
An apartment house, no (%) ^3^		499	86 (30.8)	66 (29.6)	0.986	0.01	483	49 (40.8)	136 (35.6)	0.354	0.05	152 (29.9)	185 (36.9)	0.022
Financial difficulty, no (%) ^3^		503	59 (20.9)	40 (18.1)	0.498	0.04	481	17 (14.8)	72 (19.7)	0.298	0.05	99 (19.7)	89 (18.5)	0.697
Cognitive and mental statuses (3)														
Cognitive impairment (MMSE ≤ 23), no (%) ^3^		511	49 (17.1)	42 (18.7)	0.739	0.02	500	14 (11.6)	57 (15.0)	0.422	0.04	91 (17.8)	71 (14.2)	0.139
WHO5 score, mean (SD) ^2^		504	18.2 (5.3)	18.2 (5.1)	0.900	0.00	491	18.5 (4.9)	18.5 (5.1)	0.951	0.00	18.2 (5.2)	18.5 (5.1)	0.395
GDS15, mean (SD) ^2^		504	3.3 (2.8)	3.6 (3.0)	0.377	0.04	494	2.9 (2.5)	3.2 (2.6)	0.221	0.06	3.5 (2.9)	3.2 (2.6)	0.249
Pysical functions (7)														
IADL score, mean (SD) ^2^		512	4.8 (0.4)	4.8 (0.5)	0.929	0.00	502	4.9 (0.4)	4.8 (0.5)	0.122	0.07	4.8 (0.4)	4.8 (0.5)	0.483
Frailty score, mean (SD) ^2^		502	1.4 (0.9)	1.5 (1.1)	0.353	0.04	494	1.4 (1.0)	1.4 (0.9)	0.583	0.02	1.5 (1.0)	1.4 (0.9)	0.797
BMI, mean (SD) ^1^		512	23.8 (2.6)	23.3 (3.3)	0.098	0.15	503	23.1 (3.0)	22.8 (3.2)	0.242	0.12	23.6 (2.9)	22.9 (3.2)	<0.001
Timed Up & Go (sec), mean (SD) ^2^		505	11.4 (2.7)	11.5 (2.6)	0.570	0.03	497	11.2 (2.7)	11.8 (3.3)	0.134	0.07	11.4 (2.7)	11.6 (3.2)	0.854
5m walking (sec), mean (SD) ^2^		510	5.4 (1.1)	5.6 (1.7)	0.878	0.01	499	5.3 (1.1)	5.7 (1.7)	0.012	0.11	5.5 (1.4)	5.6 (1.6)	0.735
Hand grip (kg), mean (SD) ^2^		506	27.2 (4.9)	27.4 (5.0)	0.609	0.02	497	18.5 (3.3)	18.0 (3.3)	0.092	0.08	27.3 (4.9)	18.1 (3.3)	<0.001
One-leg standing duration (sec), mean (SD) ^2^		501	23.2 (29.2)	21.5 (27.7)	0.834	0.01	493	21.5 (23.6)	20.2 (24.7)	0.278	0.05	22.4 (28.5)	20.5 (24.4)	0.842
Disease histories (6)														
Gastrointestinal disease, no (%) ^3^		511	171 (59.8)	137 (60.9)	0.872	0.01	502	72 (60.0)	227 (59.4)	0.996	0.01	308 (60.3)	299 (59.6)	0.867
Heart disease, no (%) ^3^		511	82 (28.7)	79 (35.1)	0.144	0.07	499	26 (22.0)	78 (20.5)	0.814	0.02	161 (31.5)	104 (20.8)	<0.001
Diabetes, no (%) ^3^		507	38 (13.4)	44 (19.6)	0.077	0.08	494	11 (9.2)	45 (12.0)	0.509	0.04	82 (16.2)	56 (11.3)	0.034
Cancers, no (%) ^3^		508	70 (24.5)	66 (29.7)	0.220	0.06	497	13 (11.0)	60 (15.8)	0.254	0.06	136 (26.8)	73 (14.7)	<0.001
Renal disease, no (%) ^3^		510	25 (8.7)	31 (13.8)	0.092	0.08	497	7 (5.8)	35 (9.3)	0.320	0.05	56 (11.0)	42 (8.5)	0.212
Hyperuricemia, no (%) ^3^		501	53 (18.9)	27 (12.3)	0.061	0.09	487	7 (6.2)	18 (4.8)	0.734	0.03	80 (16.0)	25 (5.1)	<0.001
Biomarkers in plasma (14)														
HDLC (mg/dL), mean (SD) ^2^		512	58.8 (14.7)	51.9 (13.6)	<0.001	0.24	503	66.9 (15.3)	65.3 (15.6)	0.367	0.04	55.8 (14.6)	65.7 (15.6)	<0.001
TCHO (mg/dL), mean (SD) ^2^		512	192.4 (31.1)	184.9 (31.7)	0.007	0.12	503	210.5 (30.5)	211.2 (34.1)	0.934	0.00	189.1 (31.5)	211.0 (33.3)	<0.001
LDLC (mg/dL), mean (SD) ^2^		512	106.4 (26.3)	105.6 (26.9)	0.735	0.01	503	114.8 (27.1)	117.1 (28.7)	0.494	0.03	106.0 (26.6)	116.5 (28.3)	<0.001
eGFRcreat (mL/min/1.73m2), mean (SD) ^2^		512	58.8 (14.3)	56.8 (15.1)	0.091	0.07	503	61.1 (15.6)	59.7 (13.8)	0.495	0.03	57.9 (14.7)	60.0 (14.3)	0.025
ALB (g/dL), mean (SD) ^2^		512	4.1 (0.3)	4.2 (0.3)	0.624	0.02	503	4.2 (0.3)	4.2 (0.3)	0.842	0.01	4.2 (0.3)	4.2 (0.3)	0.202
UA (mg/dL), mean (SD) ^2^		512	6.0 (1.3)	5.8 (1.2)	0.145	0.06	503	5.1 (1.2)	5.1 (1.2)	0.542	0.03	5.9 (1.3)	5.1 (1.2)	<0.001
AST (U/L), mean (SD) ^2^		512	24.4 (8.9)	23.9 (9.9)	0.260	0.05	503	24.0 (6.3)	25.8 (27.7)	0.973	0.00	24.2 (9.3)	25.4 (24.3)	0.192
ALT (U/L), mean (SD) ^2^		512	17.6 (8.8)	17.8 (10.9)	0.746	0.01	503	16.4 (5.4)	18.6 (38.1)	0.476	0.03	17.7 (9.8)	18.1 (33.3)	0.148
ALP (U/L), mean (SD) ^2^		512	229.3 (66.0)	245.0 (75.3)	0.012	0.11	503	233.7 (72.6)	239.9 (75.7)	0.447	0.03	236.2 (70.6)	238.4 (75.0)	0.762
gGTP (U/L), mean (SD) ^2^		512	35.9 (41.5)	25.1 (22.3)	<0.001	0.24	503	23.7 (40.9)	22.9 (16.7)	0.741	0.01	31.1 (34.8)	23.0 (24.7)	<0.001
CHE (U/L), mean (SD) ^2^		512	283.4 (59.7)	278.4 (59.8)	0.461	0.03	503	305.8 (57.0)	312.0 (66.8)	0.570	0.03	281.2 (59.8)	310.5 (64.6)	<0.001
LDH (U/L), mean (SD) ^2^		512	194.4 (37.2)	191.8 (40.1)	0.388	0.04	503	211.1 (38.1)	207.1 (41.2)	0.244	0.05	193.3 (38.5)	208.1 (40.5)	<0.001
HBA1c_NGSP (%), mean (SD) ^2^		512	5.9 (0.6)	6.0 (0.7)	0.047	0.09	503	5.9 (0.4)	6.0 (0.6)	0.826	0.01	6.0 (0.6)	6.0 (0.6)	0.635
GDF15, mean (SD) ^2^		512	1854.4 (859.0)	1900.0 (909.0)	0.715	0.02	503	1431.8 (497.9)	1530.2 (521.5)	0.064	0.08	1874.6 (880.7)	1506.7 (517.2)	<0.001
Genetic factors (3)														
*ALDH2* rs671 (p.E504K ), minor allele frequency ^3^		512	0.139	0.497	<0.001	0.39	503	0.103	0.281	<0.001	0.18	0.297	0.239	0.004
*ADH1B* rs1229984 (p.H48R), major allele frequency ^3^		512	0.777	0.782	0.460	0.03	503	0.747	0.797	0.125	0.05	0.780	0.785	0.632

^1^: *t*-test, ^2^: wilcoxon rank-test, ^3^: chi-square test ^4^: effect sizes for *t*-test, wilcoxon rank-test, chi-square were calcurated by cohen.d, wilcoxon effect size, and phai factor, respectively.

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
