# Peer review of "ALDH2 p.E504K Variation and Sex Are Major Factors Associated with Current and Quitting Alcohol Drinking in Japanese Oldest Old"

_genes, 2021, doi:10.3390/genes12060799_

Round 1
Reviewer 1 Report
This study is interesting where authors investigated factors associated with current drinkers and previous drinkers who have now quit in the Japanese oldest-old cohort of individuals ages between 85 and 90. I like that they used a large sample size of around 1015 individuals. Overall, they prepared a questionnaire with around 41 factors and performed multivariate regression logistic analysis for current drinkers indicating that the rs671 genotype was the most significant factor in Oldest Old men and women. Also, they find out that ALDH2 p.E504K variation is one of the major factors associated with current drinkers and drinking quitters in the Japanese Oldest Old samples. Their study design is really good. However, few issues are mentioned below –
Authors mentioned in the discussion at Lines 318 to 322 that in the previous study, The ADH1B p.H48R variation was positively associated with alcohol dependency; however, the ALDH2 p.E504K variation was negatively associated with alcohol dependency, and they found the same for ALDH2 p.E504K variation in Japanese Oldest Old. This study cannot be considered a novel study but a confirmatory study in different populations. This is a piece of important information that authors should put in the Introduction rather than putting it as a passing comment in Discussions.
In the Method and Materials section, at lines 110 to 113, the Authors mentioned that the genotypes of 0.65M SNVs were determined using the Infinium Asian Screening Array-24 v1.0 BeadChip kit, according to the manufacturer’s standard protocol. However, the authors did not mention what they used for SNV calling. The authors are recommended to provide Citations for tools they used for SNV calling and genotyping.
Authors should also clearly mention if the genotype data generated were phased genotypes or phased separately. In addition, authors should more clearly mention the parameters they used to determine the Imputation quality they performed. The authors mentioned that SNVs were screened using combination filters for imputation reliability filter (info ≥ 0.5). What combination filters authors are mentioning here? Explain those filters in more detail. Also, I am confused about understanding (info ≥ 0.5). Is it a kind of Probability score? The authors should elaborate on this.
In lines 136-137, the Authors mentioned that they used multiple statistical tests (t-test, Wilcoxon rank-sum test, or chi-square test) for evaluating the difference in baseline data. However, I did not find anywhere where authors used t-test. It is recommended that authors mention in these lines where these tests were used, which will increase the readability of the paragraph.
Reviewer 2 Report
It is an interesting study on genetic and factors on drinking behaviour among the older Japanese cohort. The study has very good data but the presentation could be improved.
In table 1, an overall comparison for both sexes could be provided. Also effect size estimates to check with the multivariate analysis could be provided.
It is not clear if this is a case-control GWAS and there is no mention of any covariates in the analysis. It would be good to include some basic covariates such as age and sex in the model. GWAS may not be sufficiently powered to identify other SNVs. Sample size limitation should be mentioned.
Avoid description of the imputation methods in the results section.
Section 3.5 could follow 3.3.
The two sets of LASSO analyses could be presented together.
Unlike the GWAS, it is not clear why the LASSO analyses are presented separately for both sexes. I think combined analysis for both sexes could be presented in the main text. And a supplementary analysis involving separate sexes could be presented in the supplementary text.
The multivariate analysis to find factors associated with drinking behaviours may not be very useful. Some of these factors may be due to reverse causation, for example, the blood biomarkers and cognitive status etc. could be affected due to drinking behaviour but not causing drinking behaviour.
There is not much discussion about the multivariate analysis in the discussion.
Reviewer 3 Report
The paper is overall well written and the topic is interesting for its thematic area. Indeed, the results obtained by the authors fit very well with the objectives and methodologies used. Overall, the data showed are robust and also supported by statistical analysis.
